# Quota System in Japanese Politics, Healthcare, and Education: Women's Rights and Equality

**Eisuke Nakazawa** [1] and **Akira Akabayashi** [1,2,*]

1    Department of Biomedical Ethics, University of Tokyo Faculty of Medicine, Tokyo 113-0033, Japan
2    Division of Medical Ethics, New York University School of Medicine, New York, NY 10016, USA
*    Correspondence: akira.akabayashi@gmail.com or akirasan@m.u-tokyo.ac.jp; Tel.: +81-35841-3511; Fax: +81-35841-3319

**Abstract:** The World Economic Forum's Gender Gap Index shows the serious nature of Japan's gender gap. The gender gap with respect to political and economic participation is obvious. The percentage of women in the Diet (the national parliament) is significantly low, and few women hold management positions. Although not shown in the Gender Gap Index, there are hidden gender inequalities in education and health care in Japan. These gender inequalities are largely due to the remnants of Japan's traditional family culture and customs. In order to empower Japanese women from the confines of the family and community, drastic measures guided by the principles of gender equality, diversity, and inclusion are needed. A quota system is one of the most important strategies to achieve this, and the time has come for Japan to seriously discuss the introduction of a quota system in politics, corporate culture, and university entrance examinations.

**Keywords:** gender equality; education; politics; quota system

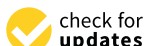



## 1. Introduction: Gender Gap Index

On 13 July 2022, the World Economic Forum ranked Japan 116th (0.650) of 146 countries in the Gender Gap Index, which weighs data in the areas of economy, education, health, and politics [1]. The Gender Gap Index value is rated on a scale from 0 to 1. The closer the index value is to 1, the higher the gender equality. North European countries occupy the top three positions in the index, with Iceland (0.908), Finland (0.860), and Norway (0.845) in first, second, and third place, respectively [1]. A detailed look at Japan's situation indicates that the index values for economic participation, political participation, education, and health are 0.564, 0.061, 1, and 0.973, respectively. While these values indicate that gender equality has been achieved in the areas of education and health, it is evident that not enough has been done in the areas of economic and political participation.

Inspired by these findings of the Gender Gap Index, this op-ed article reports on the depth of Japan's gender gap and considers what may be necessary to create gender equality in Japanese society. "Gender equality" is a central term in feminist theory and theorists have debated its conceptualization for several generations [2]. Consequently, different conceptualizations of "gender equality" reflect the broader history of feminist disputes of which it is a part. However, the concept can sometimes also carry an empty meaning. In brief, "gender equality" can be divided into "formal equality" and "substantive equality". "Formal equality" refers to equality guided by the principle of gender parity, such as equal numbers of male and female parliamentarians. However, gender parity is nothing more than a compensatory concept of equality with males as the norm. Nevertheless, formal equality can be seen as a first step toward substantive equality, a more positive and diverse type of equality that takes into account the differences across all genders [3]. This paper treats "gender equality" as formal equality and thus keeps gender parity in mind. This strategy is a methodological measure to approach substantive equality beyond that.

## 2. Women's Participation in Politics and Economy

As of 28 April 2021, the percentage of women in the House of Representatives was 9.9%; the percentage of women in the House of Councilors as of 23 May 2022 was 23.0%; and the total percentage of women in the National Assembly was 14.4% (as of 23 May 2022) [4]. The percentage of women in the national Diet (lower or single house) is significantly lower than that in the G7 countries: 39.5% in France, 33.9% in the United Kingdom, 31.5% in Germany, 29.6% in Canada, and 27.3% in the United States of America [4]. In the past, Japan produced historically outstanding female politicians such as Raicho Hiratsuka and Fusae Ichikawa. One female politician active in recent years is Takako Doi, who in 1989 became the first woman to be nominated to head the Cabinet of the House of Councilors. (Due to the superiority of the House of Representatives, it was a man, Toshiki Kaifu, who was ultimately nominated by the House of Representatives, and became Prime Minister.) Takako Doi further became the first female Speaker of the House of Representatives in 1993. Among female politicians of the current generation, the successes of Yuriko Koike, who became the first female governor of Tokyo in 2016, are well-known. Sanae Takaichi and Seiko Noda, who ran for the Liberal Democratic Party presidency, a ruling party in Japan, in the election that would eventually choose Japan's Prime Minister, have caught the eye of the public. However, most famous politicians are men, indicating an excessive gender imbalance in politics today.

When it comes to economic participation, there is concern that women in Japan have few opportunities to play a leadership role in economic participation. The number of women in management positions in Japan remains low. According to a survey conducted by Teikoku Databank Corporation (survey period: 15–31 July 2021), the average percentage of female managers (section manager equivalent or higher) in 10,992 Japanese companies was only 8.9% (still a record high in similar annual surveys). Only 8.6% of companies had achieved the government's target of 30% women in management positions [5]. The United Nations Development Programme's Gender Inequality Index (2021) shows that Japan ranks 22nd out of 170 countries in terms of index value [6]. However, a closer look at the content of the index shows that the labor force participation rate is relatively low at 53.3%, ranking 85th among all countries [6]. The difference from the male labor force participation rate of 71.0% is also quite large, indicating that a gender gap still exists in Japan in terms of economic participation. It is clear that the burden of childbirth, childcare, and housework is disproportionately placed on women, and this is the primary reason why the number of women in management positions has not increased. There still remains pre-World War II norms in Japan: "Women's happiness is to get married and have children, and women's work is housework and childcare, and caring for their parents is also women's work. It is the role of men to work in society, and women should support them". The Gender Inequality Index tends to be higher in developed countries because of their focus on human development. The relatively high Gender Inequality Index and low Gender Gap Index indicate a significant gender gap within a country, especially in terms of economic status. It can be inferred that the overburden of unpaid domestic work for women in Japan is behind these problems. The low number of women in management positions in Japan, as indicated by the Teikoku Databank Corporation, is certainly rooted in the fact that women's career paths in socioeconomic activities are hampered by the persistence of traditional values. The introduction of quotas for women in management positions makes sense in order to improve this gender imbalance as much as possible. Increasing the number of female managers through the introduction of a quota system will lead to a change in social structure. Drastic policies are needed to drastically change Japan's traditional norms regarding women and the family.

## 3. Gender Gap in Healthcare

According to the Gender Gap Index, Japan appears to have achieved gender equality in the areas of education (1.000) and health (0.973), but not in economic and political participation. However, hidden gender inequalities are inherent in Japanese society, both

in healthcare and in education. This does not mean that there are no obvious and serious gender differences in patients' access to medical care. However, the burden on women in terms of reproductive health care must be taken into account. A woman's freedom to have an abortion is not recognized by Japanese law, and the crime of abortion under the Penal Code applies. Nevertheless, the Maternal Protection Law allows abortion of pregnancies resulting from rape and for economic reasons [7]. Broadly interpreting this "economic reason", Japan's current situation is that a woman's freedom to abort is endorsed "through the back door". The discrepancy between clear provision in the law and the actual implementation of the law is not a desirable state of affairs. If abortion is to be implemented socially, its basic legal and ethical principles should be thoroughly discussed in the Diet. Otherwise, women who are forced to seek abortion are placed in a legally precarious position. This is a violation of women's rights.

In addition, the physical, mental, economic, and social burdens of infertility treatment are concentrated on women. Infertility treatment is a serious medical procedure, aside from its physical and emotional severity. Although the financial burden is shared with the spouse, it is still a huge burden for women. To reduce this financial burden, from April 2022, "general infertility treatment" such as artificial insemination and "assisted reproductive medical treatment" (in vitro fertilization and intracytoplasmic sperm injection) will be covered by public insurance [8]. These policies are effective in terms of respecting the rights of couples to have children on a socially equal basis. At the same time, however, there is a fear that public insurance coverage for some infertility treatments may lead to a disregard of women's rights to not have children. Japan is a cohesive, family-oriented society, and there is still a strong tendency for women to believe that their role in the home is to bear and raise children. This means that the patriarchal system of World War II is still in place in Japan today. During World War II, it was considered a woman's duty to give birth to and raise a fine boy who would protect the nation and the family. In the postwar era, as Westernization and globalization progressed, the Japanese patriarchal system gradually fell out of prominence. However, its remnants still linger in Japanese society like a shadow. In the history of women's rights, which have been wedged within the family in this way, consideration of a woman's right to not have children is of great importance today.

Furthermore, looking at the medical field, the low number of female physicians in Japan is striking. According to OECD statistics released in July 2022, the percentage of female physicians in Japan was 22.71% in 2020, the lowest among OECD member countries. Moreover, Japan stands out as the country with the lowest ratio of female doctors among OECD member countries, with Latvia topping the list at 73.94% and figures of more than 40% being reached in other countries. Structural problems, including the entrance examinations of medical schools in Japan, are reasons for this situation. A corruption case uncovered in August 2018, in which Tokyo Medical University gave preferential treatment to children of Ministry of Education, Culture, Sports, Science and Technology bureaucrats in its entrance examinations, led to confirmation of the lack of suitability of entrance examinations, including discrimination against women and age discrimination, at nine medical schools, which also included Tokyo Medical University [9]. At the very least, Japan's gender inequality will not be resolved without correcting these inequalities, particularly the gender balance in education. Concerning this, it is important to examine the status of gender inequality with regard to education in general in Japan.

## 4. The Gender Gap in Education: The Case of the University of Tokyo

An extensive observation of the situation reveals that educational opportunities for men and women in Japan are unequal. For example, the ratio of female students at the University of Tokyo is 2802/13,962 (20.1%) for undergraduates, 1778/7218 (24.6%) for master's students, and 1816/6123 (29.7%) for doctoral courses [10]. The University of Tokyo is known as the most highly ranked university in Japan, and many of its graduates are candidates for bureaucratic or corporate management positions. The gender imbalance in enrollment at the University of Tokyo may lead to a low percentage of female managers

in the corporate sector, and a low percentage of female bureaucrats and politicians in the future. Norio Matsuki, a member of the University of Tokyo's Board of Trustees, attributes this to the following:

One of the main reasons for the difference in the male–female ratio is that fewer girls aim to enter the University of Tokyo when they apply for university entrance exams. "Boys go to good universities, get good jobs, and take over the family. In contrast, girls protect the family and support the boys". Such a gender role division may be still deeply rooted in Japanese society. It seems that many parents want their boys to move to Tokyo, but want to keep the girls close at hand, and are concerned about having them live alone in faraway Tokyo [11].

Matsuki's point here is that Japan's patriarchal system also casts a dark shadow on the context of education. The fact that such a gender gap can be observed even in the country's fundamental system of education indicates that Japan is still lacking in its efforts toward gender equality.

The University of Tokyo promotes an environment that facilitates student life for female students, especially those from areas far from Tokyo, by providing housing assistance and scholarships for female students. In June 2022, the University of Tokyo Statement on Diversity and Inclusion was presented, stating it will work to eliminate the gender gap and other inequalities, and promote diversity and inclusion as a university [12]. These efforts are expected to restore the gender balance at the University of Tokyo to an appropriate level, as well as change the social structure of Japan, which has created the gender gap. Furthermore, reform of university entrance examinations is also underway in some areas. Instead of the usual paper-based entrance examination, a system to recruit a more diverse student body has been adopted in the form of a recommendation-based entrance examination. University entrance examinations are of paramount importance to any university, especially in Japanese society, which places a high value on university admission history. The University of Tokyo's expansion of its admission examination system is commendable as it opens the door to a diverse range of students.

However, both support for female students and entrance examinations based on admission qualifications with recommendations are still too small in scale as measures to close the gender gap in education that stems from Japan's traditional social structure. There is still room for further study, including the introduction of more drastic measures to foster female leaders who will lead society in the future.

## 5. Glass Ceiling and Quota System

As indicated above, Japanese women are restricted in their future planning by the remnants of the family system in Japan. This structure has contributed to the low number of female politicians and corporate managers. Hillary Clinton, who lost the U.S. presidential election, said in her speech, "To all the women and especially the young women who put their faith in this campaign and in me, I want you to know that nothing has made me prouder than to be your champion, now, I know, I know we have still not shattered that highest and hardest glass ceiling, but someday, someone will" [13]. The glass ceiling that American women have been unable to break through also exists for Japanese women. Moreover, these barriers bind Japanese women from birth, like heavy chains tied around their legs.

An effective solution to this situation is the introduction of a quota system, in which a certain number of candidates for parliamentary seats must be women. Originating in Norway, this system has already spread to countries around the world [14] and is considered to play an important role in realizing a gender-equal society. Japan may begin to consider a quota system with respect to politics (Diet and local assembly members, and bureaucrats), economics (general business), and education (university admissions).

A study conducted in South Korea found that the introduction of a quota system for university faculty members showed "gender quotas have a positive effect on female faculty representation at all levels of tenured and tenure-track professorship but not for leadership

and higher administrative positions such as Dean, Provost, and President" [15]. In addition, according to Akala (2019), gender inequality in education has not been corrected despite the introduction of gender quota systems in higher education in South Africa and Kenya [16]. One direction that can be taken from these previous studies is that while the introduction of quota systems can, of course, be expected to have some effect in closing the gender gap, it is not a definitive solution, but only a supplemental one.

The effect may be limited, but it is time for Japan to consider boldly introducing the quota system not only in politics but also in economics and education. This is because, while grassroots bottom-up efforts may be necessary to emancipate women from the family and community traditions that have been ingrained in Japanese tradition, they are more effective when combined with the introduction of top-down solutions guided by a philosophy. Starting with the Diet and local assembly members, the quota system should be expanded to include bureaucrats and faculty members of national universities and be applied when admitting students at these national universities in Japan. In addition, policies should be implemented to provide incentives for private companies and universities to ensure gender equality.

Looking at public opinion in Japan, the quota system is criticized for being a form of reverse discrimination, stifling free competition, and preventing the best and brightest from playing an active role in society. Another criticism is that women are not the only disadvantaged minority group; it is therefore unequal to give preferential treatment to women, without considering other minorities. This may be true, but a good strategy to eliminate all such inequalities in the future would be to start with gender inequality.

Of course, this does not mean that other forms of discrimination should be allowed to go unchecked. No one should be discriminated against based on family origin or ethnicity (there are many different ethnicities in Japan), and discrimination on the basis of disability is also unacceptable. Discrimination against lesbian, gay, bisexual, and transgender people is also a serious issue and affirmative action on this type of discrimination must be promoted. Nevertheless, it is reasonable to start with the resolution of gender inequality with regard to implementation of a quota system. Since nearly half of the population is female, the resolution of gender inequality is superior from a utilitarian perspective. Furthermore, designing each minority-specific quota through a quota system becomes more difficult the larger the number of minorities targeted. Since it takes a certain amount of time to address these difficulties, it is more feasible to first aim at eliminating gender inequality.

## 6. Scope of the Quota System and Culture

Given the above, we must consider to what extent the quota system should be expanded. Politics, economics, and education are areas where the quota system should be actively considered, as they are highly public. It must be noted that with regard to culture and religion, there are many examples of societies formed exclusively by men in Japan's traditional performing arts and culture. Sumo is both a sport and a traditional performance art [17] but, traditionally, only men are allowed in the ring [18,19]. Traditional performing arts such as Noh and Kabuki also only permit males to participate. The same is true of religious practices, with women being forbidden to climb some mountains due to ancient mountain beliefs.

While these traditional cultures and customs are subject to occasional criticism, we need to be more cautious about introducing a quota system into our culture. This would require broad public debate. The emperor system in Japan is more a cultural than political system and, in recent years, there has been some discussion about female emperors in order to stabilize succession to the throne. Perhaps culture is slowly changing in accordance with the ethos of the times.

## 7. Conclusions

The World Economic Forum's Gender Gap Index shows the serious nature of Japan's gender gap. While the gender gap with respect to political and economic participation is

obvious, the gender gap in terms of education and healthcare remains hidden. The remnants of the prewar patriarchal system still linger in the background of gender inequality in Japan. The vestiges of this system are deeply ingrained in community and family values. Japanese women carry the burden of domestic work in the family and need to be freed from this unpaid labor. The systematic introduction of policies driven by ideals will be necessary to change the social climate that has become so ingrained in society. The introduction of a quota system is one of the leading strategies to combat this. It is time for Japan to begin serious discussions on the introduction of a quota system. First, lawmakers and bureaucrats should begin with discussing the introduction of a quota system. Furthermore, if education is the pillar of the nation that will shape the future of society, then there should be a serious discussion about introducing a quota system for university entrance examinations in order to drastically improve the situation of women in Japan.

**Author Contributions:** Conceptualization, E.N. and A.A.; writing—original draft preparation, E.N. and A.A.; writing—review and editing, E.N. and A.A.; project administration, A.A. All authors have read and agreed to the published version of the manuscript.

**Funding:** This research received no external funding.

**Institutional Review Board Statement:** Not applicable.

**Informed Consent Statement:** Not applicable.

**Data Availability Statement:** Not applicable.

**Conflicts of Interest:** The authors declare no conflict of interest.

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
