# Peer review of "Quota System in Japanese Politics, Healthcare, and Education: Women’s Rights and Equality"

_sexes, doi:10.3390/sexes3040037_

Round 1

Reviewer 1 Report

This is an important and well written article which examine Japan's gender gap and the possible use of a quota system in government positions and the education system to address gender inequalities.

A few aspects of the article could be further clarified.

-- The discussion of economic participation, beginning on line 53, could be a separate paragraph. Also, it seems that the focus is only on women in management positions. It would be useful to provide some discussion of why this is the only indicator discussed, and how Japan fared on other indicators of economic participation.

A focus on women in management positions also raises issues in terms of gender and class intersectionalities. Will the interests of all women be served if some women are inducted into management positions? What other occupations are women involved in and how can their position in these occupations be improved?

-- This sentence could be further clarified [lines 89-92]: 'This is a remnant of the patriarchal system that existed until the time before World War II, and up until the first half of the 20th century, it was considered a woman's duty to give birth to and raise a fine boy who would protect the nation and the family.'

This sentence seems to suggest that the 'patriarchal system' did not exist after World War II. Also, the term 'remnant' [also mentioned on line 130] seems to suggest that patriarchy has become less visible and weaker in Japan. Might be useful instead to contrast older forms of patriarchy with continuing forms of patriarchy.

-- In lines 119-127, it is not clear which parts are a specific quote from Norio Matsuki and which parts are the authors' words. If the entire paragraph is a quote, then the quotes within the paragraph need to be removed.

-- On line 151, this sentence could be deleted: 'Of course, these criticisms are misguided.' While this is an opinion article, this sentence seems out of place here, since why the criticisms are misguided is mentioned in the next paragraph. Maybe this paragraph [line 149-155] can focus on criticisms of the quota system, since the next paragraph outlines reasons to support the quota system.

Also, the meaning of: 'a serious and larger complication at hand' [line 155] is not clear. Maybe this could be deleted, or the sentence of which it is a part could be revised.

-- What is the relationship between affirmative action and quota system according to the authors? Are these two the same terms, or is there a difference between them?

Delete 'in affirmative action' on line 162, since this makes the sentence unclear?

-- The last sentence of the article is too abrupt [pp.193-194. Maybe it can be revised, or an additional concluding sentence added after it.

Author Response

Thank you very much for reading our communication letter so carefully and for your very useful comments. We believe we could have improved the manuscript by discussing your points. Thank you very much.

-- The discussion of economic participation, beginning on line 53, could be a separate paragraph. Also, it seems that the focus is only on women in management positions. It would be useful to provide some discussion of why this is the only indicator discussed, and how Japan fared on other indicators of economic participation.

A focus on women in management positions also raises issues in terms of gender and class intersectionalities. Will the interests of all women be served if some women are inducted into management positions? What other occupations are women involved in and how can their position in these occupations be improved?

Thank you very much. First, we have separated the paragraphs for the part describing the economic aspects. We have also made the following additions regarding women and labor and increasing the number of women in management positions.

When it comes to economic participation, there is concern that women in Japan have few opportunities to play a leadership role in economic participation. The number of women in management positions in Japan remains low. According to a survey conducted by Teikoku Databank Corporation (survey period: July 15-31, 2021), the average percentage of female managers (section manager equivalent or higher) in 10,992 Japanese companies was only 8.9% (still a record high in similar annual surveys). Only 8.6% of companies had achieved the government’s target of 30% women in management positions [3]. The United Nations Development Programme’s Gender Inequality Index (2021) shows that Japan ranks 22nd out of 170 countries in terms of index value [4]. However, a closer look at the content of the index shows that the labor force participation rate is relatively low at 53.3%, ranking 85th among all countries [4]. The difference from the male labor force participation rate of 71.0% is also quite large, indicating that a gender gap still exists in Japan in terms of economic participation. It is clear that the burden of childbirth, childcare, and housework is disproportionately placed on women, and this is the primary reason why the number of women in management positions has not increased. There still remains pre-World War II norms in Japan: “Women’s happiness is to get married and have children, and women's work is housework and childcare, and caring for their parents is also women’s work. It is the role of men to work in society, and women should support them.” The Gender Inequality Index tends to be higher in developed countries because of their focus on human development. The relatively high Gender Inequality Index and low Gender Gap Index indicate a significant gender gap within a country, especially in terms of economic status. It can be inferred that the overburden of unpaid domestic work for women in Japan is behind these problems. The low number of women in management positions in Japan, as indicated by the Teikoku Databank Corporation, is certainly rooted in the fact that women’s career paths in socioeconomic activities are hampered by the persistence of traditional values. The introduction of quotas for women in management positions makes sense in order to improve this gender imbalance as much as possible. Increasing the number of female managers through the introduction of a quota system will lead to a change in social structure. Drastic policies are needed to drastically change Japan’s traditional norms regarding women and the family.

-- This sentence could be further clarified [lines 89-92]: 'This is a remnant of the patriarchal system that existed until the time before World War II, and up until the first half of the 20th century, it was considered a woman's duty to give birth to and raise a fine boy who would protect the nation and the family.'

This sentence seems to suggest that the 'patriarchal system' did not exist after World War II. Also, the term 'remnant' [also mentioned on line 130] seems to suggest that patriarchy has become less visible and weaker in Japan. Might be useful instead to contrast older forms of patriarchy with continuing forms of patriarchy.

Thank you very much for making exactly the right point, and I sincerely appreciate it. I do not think that the word "remnant" was appropriate with regard to the Japanese patriarchal system. Rather, we would like to emphasize that the patriarchal system still haunts Japan like a ghost, from which we still cannot escape. We have changed the sentence to the following.

This means that the patriarchal system of World War II is still in place in Japan today. During World War II, it was considered a woman’s duty to give birth to and raise a fine boy who would protect the nation and the family. In the postwar era, as Westernization and globalization progressed, the Japanese patriarchal system gradually fell out of prominence. However, its remnants still linger in Japanese society like a shadow.

-- In lines 119-127, it is not clear which parts are a specific quote from Norio Matsuki and which parts are the authors' words. If the entire paragraph is a quote, then the quotes within the paragraph need to be removed.

I am very sorry for the confusion. All paragraphs here are quotes. We have clarified this and inserted comments that are in line with the quotes.

Matsuki’s point here is that Japan’s patriarchal system also casts a dark shadow on the context of education. The fact that such a gender gap can be observed even in the country’s fundamental system of education indicates that Japan is still lacking in its efforts toward gender equity.

-- On line 151, this sentence could be deleted: 'Of course, these criticisms are misguided.' While this is an opinion article, this sentence seems out of place here, since why the criticisms are misguided is mentioned in the next paragraph. Maybe this paragraph [line 149-155] can focus on criticisms of the quota system, since the next paragraph outlines reasons to support the quota system.

Also, the meaning of: 'a serious and larger complication at hand' [line 155] is not clear. Maybe this could be deleted, or the sentence of which it is a part could be revised.

Thank you for pointing that out. We have removed it.

-- What is the relationship between affirmative action and quota system according to the authors? Are these two the same terms, or is there a difference between them?

Delete 'in affirmative action' on line 162, since this makes the sentence unclear?

Thank you very much for pointing this out. We have removed "in affirmative action." We believe that the quota system is a type of affirmative action.

-- The last sentence of the article is too abrupt [pp.193-194. Maybe it can be revised, or an additional concluding sentence added after it.

Very sorry, this is my mistake. I have deleted it.

Reviewer 2 Report

Dear authors, I read your manuscript. I found the topic relevant; despite this, I did not find your argumentations convincing. Here I provide some suggestions for improving your work.

The abstract structure is quite fragmented so it is not clear.

Furthermore, the meaning of “gender gap” requires deeper explanation as it is not clear whether the authors consider it related to pay gap, representation, work-life balance, and so on. This is evident when the authors state that “According to the Gender Gap Index, Japan appears to have achieved gender equality in the areas of education (1.000) and health (0.973), unlike in economic and political participation”, so it is not clear what they refer in terms of “gender equality”, also given that they introduce topic such as access to medical care, reproductive health, female professionals, education (which is addressed in sole 4 lines). In this sense, how the authors link the different topic is not clear.

Furthermore, paragraph 3 is not structured enough. Specifically, the title is “The Gender Gap in Education: The Case of the University of Tokyo”. Despite this, neither the educational situation in Japan, nor the specific case of the University of Tokyo are deepened

Also, it puzzles me how authors derive general statements from single events (e.g. “The glass ceiling that American women have been unable to break” from Hilary Clinton loosing elections).

The authors present the quota system as an effective solution, despite not providing any evidence of that in terms of references. Further, the solution presented is too general and not adequately framed (the authors suggest to apply it to politics, economics and education, with little discussion).

In addition, authors suggest cultural changes in Japan, while previous discussion was highly rooted on existing inequalities, thus this need additional explanation.

Conclusions partially replicate the introduction and do not really provide concluding comments. I suggest the authors to revrite them.

Moreover, the reference need to be revised to include more relevant and international literature on the topic.

I suggest the authors to focus on a specific topic and to better build their discussion around it. It seems that in this version there are too topic which are not adequately linked and the final message is missing.

Minor issues: In the abstract, the term DIET is not defined.

Author Response

Thank you very much for reading our communication letter so carefully and for your very useful.

We have substantially rewritten the Abstract, Section 3, and Conclusions per your suggestion.

The first important point you made to me is about the structure of the abstract and conclusion. You wrote: “The abstract structure is quite fragmented so it is not clear.” “Conclusions partially replicate the introduction and do not really provide concluding comments. I suggest the authors to revrite them.” We have re-worked our paper and we have revised our abstract and conclusions as follows We believe that this reorganization makes our point more clearly. We are indebted to you for this. Thank you very much.

Abstract

The World Economic Forum’s Gender Gap Index shows the serious nature of Japan’s gender gap. The gender gap with respect to political and economic participation is obvious. The percentage of women in the Diet is significantly low, and few women hold management positions. Although not shown in the Gender Gap Index, there are hidden gender inequalities in education and health care in Japan. These gender inequalities are largely due to the remnants of Japan’s traditional family culture and customs. In order to empower Japanese women from the confines of the family and community, drastic measures guided by the principles of gender equality, diversity, and inclusion are needed. A quota system is one of the most important strategies to achieve this, and the time has come for Japan to seriously discuss the introduction of a quota system in politics, corporate culture, and university entrance examinations.

Conclusion

The World Economic Forum’s Gender Gap Index shows the serious nature of Japan’s gender gap. While the gender gap with respect to political and economic participation is obvious, the gender gap with respect to education and healthcare remains hidden. The remnants of the prewar patriarchal system still linger in the background of gender inequality in Japan. The vestiges of this system are deeply ingrained in community and family values. Japanese women carry the burden of domestic work in the family and need to be freed from this unpaid labor. The systematic introduction of policies driven by ideals will be necessary to change the social climate that has become so ingrained in society. The introduction of a quota system is one of the leading strategies to combat this. It is time for Japan to begin serious discussions on the introduction of a quota system. First, lawmakers and bureaucrats should begin with discussing the introduction of a quota system. Fur-thermore, if education is the pillar of the nation that will shape the future of society, then there should be a serious discussion about introducing a quota system for university en-trance examinations in order to drastically improve the situation of women in Japan.

Then, you mentioned as follows about section 3:

Furthermore, paragraph 3 is not structured enough. Specifically, the title is “The Gender Gap in Education: The Case of the University of Tokyo”. Despite this, neither the educational situation in Japan, nor the specific case of the University of Tokyo are deepened

Here again, you are really right. We have made the following changes to the manuscript:

Matsuki’s point here is that Japan’s patriarchal system also casts a dark shadow on the context of education. The fact that such a gender gap can be observed even in the country’s fundamental system of education indicates that Japan is still lacking in its efforts toward gender equity.

              The University of Tokyo promotes an environment that facilitates student life for fe-male students, especially those from areas far from Tokyo, by providing housing assistance and scholarships for female students. In June 2022, the University of Tokyo Statement on Diversity and Inclusion was presented, stating it will work to eliminate the gender gap and other inequalities, and promote diversity and inclusion as a university [10]. These efforts are expected to restore the gender balance at the University of Tokyo to an appropriate level, as well as change the social structure of Japan, which has created the gender gap. Furthermore, reform of university entrance examinations is also underway in some areas. Instead of the usual paper-based entrance examination, a system to recruit a more diverse student body has been adopted in the form of a recommendation-based entrance examination. University entrance examinations are of paramount importance to any university, especially in Japanese society, which places a high value on university admission history. The University of Tokyo’s expansion of its admission examination system is commendable as it opens the door to a diverse range of students.

              However, both support for female students and entrance examinations based on admission qualifications with recommendations are still too small in scale as measures to close the gender gap in education that stems from Japan’s traditional social structure. There is still room for further study, including the introduction of more drastic measures to foster female leaders who will lead society in the future.

You also suggested that we discuss the quota system in an international context. Your suggestion has opened up a world of possibilities for us. We are truly and sincerely grateful. We have added the following to Section 4

A study conducted in South Korea found that the introduction of a quota system for university faculty members showed “gender quotas have a positive effect on female faculty representation at all levels of tenured and tenure-track professorship but not for leadership and higher administrative positions such as Dean, Provost, and President” [13]. In addition, according to Akala (2019), gender inequality in education has not been corrected de-spite the introduction of gender quota systems in higher education in South Africa and Kenya [14]. One direction that can be taken from these previous studies is that while the introduction of quota systems can, of course, be expected to have some effect in closing the gender gap, it is not a definitive solution, but only a supplemental one.

The effect may be limited, but it is time for Japan to consider boldly introducing the quota system not only in politics but also in economics and education. This is because, while grassroots bottom-up efforts may be necessary to emancipate women from the family and community traditions that have been ingrained in Japanese tradition, they are more effective when combined with the introduction of top-down solutions guided by a philosophy.

Minor issues: In the abstract, the term DIET is not defined.

Thank you, I clarified it as national Diet (Lower or single house).

Round 2

Reviewer 2 Report

Dear Authors,

I recognize you made and extensive work in revising the paper. I still have two main concerns to this version:

- The first refer to the term "equity", which is different from "equality" and refer to a different concept. I suggest the authors to maintain the term equality and replace equity in the text (especially in the title).

- The second is related to the concept of "gender equality" itself. Authors mainly conceive gender equality in terms of balancing the proportions of women and men in different positions and at different stages of their careers/education. I suggest them to discuss up front the meaning attributed to the term gender equality using relevant literature (you will find several studies reporting the different features of gender equality), as this can support a better comprehension of the remaining of the article.

Minor issues

In the abstract, the authors refer to Diet. Diet is not a well known institution outside Japan, so maybe they can refer to "Diet (the national parliament)" or something similar.

The ordering of the paragraphs should be checked (there are two paragraph 2).

Author Response

We would like to thank Reviewer 2 from the bottom of our hearts. Thank you so much for kindly considering our opinion article. Your insights have improved our paper.

- The first refer to the term “equity”, which is different from “equality” and refer to a different concept. I suggest the authors to maintain the term equality and replace equity in the text (especially in the title).

Thank you so much for advising. We have modified the terms accordingly.

- The second is related to the concept of "gender equality" itself. Authors mainly conceive gender equality in terms of balancing the proportions of women and men in different positions and at different stages of their careers/education. I suggest them to discuss up front the meaning attributed to the term gender equality using relevant literature (you will find several studies reporting the different features of gender equality), as this can support a better comprehension of the remaining of the article.

Thank you very much for this important recommendation. In response, we have clarified our position in the Introduction of this paper as follows:

Inspired by these findings of the Gender Gap Index, this op-ed article reports on the depth of Japan’s gender gap and considers what may be necessary to create gender equality in Japanese society. “Gender equality” is a central term in feminist theory and theorists have debated its conceptualization for several generations [2]. Consequently, different conceptualizations of “gender equality” reflect the broader history of feminist disputes of which it is a part. However, the concept can sometimes also carry an empty meaning. In brief, “gender equality” can be divided into “formal equality” and “substantive equality.” “Formal equality” refers to equality guided by the principle of gender parity, such as equal numbers of male and female parliamentarians. However, gender parity is nothing more than a compensatory concept of equality with males as the norm. Nevertheless, formal equality can be seen as a first step toward substantive equality, a more positive and diverse type of equality that takes into account the differences across all genders [3]. This paper treats “gender equality” as formal equality and thus keeps gender parity in mind. This strategy is a methodological measure to approach substantive equality beyond that.

In the abstract, the authors refer to Diet. Diet is not a well known institution outside Japan, so maybe they can refer to "Diet (the national parliament)" or something similar.

Thank you so much for pointing this out. We have revised the expression as you advised.

The ordering of the paragraphs should be checked (there are two paragraph 2).

We sincerely apologize for this oversight. We have changed the text accordingly.
